# The Australian Eye and Ear Health Survey (AEEHS): Study protocol for a population-based cross-sectional study

Richard Kha[1]*, Oonagh Macken[2], Paul Mitchell[1], Gerald Liew[1], Lisa Keay[3], Colina Waddell[4], Eleanor Yang[3], Vu Do[1], Tim Fricke[3,4], John Newall[3], Bamini Gopinath[3]

1 Centre for Vision Research, Department of Ophthalmology, The Westmead Institute for Medical Research, Westmead, Sydney, NSW, Australia, 2 Macquarie University Hearing, Faculty of Medicine, Health and Human Sciences, The Australian Hearing Hub, Macquarie University, Macquarie Park, NSW, Australia, 3 School of Optometry and Vision Science, Faculty of Science, UNSW Sydney, Sydney, NSW, Australia, 4 Brien Holden Foundation, Sydney, NSW, Australia

* rkha2425@uni.sydney.edu.au

## Abstract

### Introduction

Vision and hearing impairments are highly prevalent and have a significant impact on physical, psychological and social wellbeing. There is a need for accurate, contemporary national data on the prevalence, risk factors and impacts of vision and hearing loss in Australian adults.

### Objectives

The Australian Eye and Ear Health Survey (AEEHS) aims to determine the prevalence, risk factors and impacts of vision and hearing loss in both Aboriginal and Torres Strait Islander and non-Indigenous older adults.

### Methods and analysis

The AEEHS is a population-based cross-sectional survey which will include 5,000 participants (3250 non-Indigenous aged 50 years or older and 1750 Aboriginal and Torres Strait Islander people aged 40 years or older) from 30 sites covering urban and rural/regional geographic areas, selected using a multi-stage, random cluster sampling strategy. Questionnaires will be administered to collect data on socio-demographic, medical, ocular and ontological history. The testing battery includes assessment of blood pressure, blood sugar, anthropometry, visual acuity (presenting, unaided, pinhole and best-corrected), refraction, tonometry, slit lamp and dilated eye examination, ocular imaging including optical coherence tomography (OCT), OCT-angiography and retinal photography, and automated visual fields. Audiometry, tympanometry and video otoscopy will also be performed. The primary outcomes are age-standardised prevalence of cause-specific vision and hearing impairment. Secondary outcomes are prevalence of non-blinding eye diseases (including

**Data Availability Statement:** No datasets were generated or analysed during the current study.

**Funding:** P.M, G.L, L.K, B.G and C.W were awarded a tender by the Australian Government

Department of Health and Aged Care to conduct this research project. https://www.health.gov.au/. The funders had and will have a role in the study design, data collection and analysis, decision to publish or preparation of the manuscript.

**Competing interests:** The authors have declared that no competing interests exist.

dry eye disease), patterns in health service utilisation, universal health coverage metrics, risk factors for vision and hearing impairment, and impact on quality of life.

## Introduction

Vision and hearing impairment have been found to be highly prevalent chronic conditions that have significant impacts on health and wellbeing [1]. As such, contemporary studies to establish the prevalence, risk factors and impact of vision and hearing loss among Aboriginal and Torres Strait Islander people and non-Indigenous Australians are needed. Globally, 285 million individuals are visually impaired, of whom 39 million are blind [2]. In Australia, there are 840,000 individuals estimated to be living with vision loss, and this number is expected to exceed 1.04 million by 2030 [3]. Furthermore, there are currently 3.6 million Australians estimated to be living with hearing loss and this is projected to rise to 7.8 million by 2060 [4].

The social, economic and health system costs of vision and hearing impairment are significant. In Australia, the economic impact of vision and hearing loss is $27.6 and $33.3 billion per year, respectively [3, 5–7]. Vision and hearing loss are strongly associated with reduced quality of life, mental and physical health, as well as increased mortality and social isolation [6, 8, 9]. Over 90% of vision and hearing loss can be prevented or treated through identification of risk factors, early diagnosis and management [10, 11]. Hence, accurate age- and sex-specific data for visual and hearing impairment over time are critical.

In addition to the need for national prevalence data in Australia, there is also a need to identify population subgroups with greater burden of vision and hearing loss. Visual and hearing impairment rates have previously been estimated to be three and ten times higher among Aboriginal and Torres Strait Islander people when compared to non-Indigenous communities, respectively [12–14]. Furthermore, global studies have found that eye and ear conditions are more prevalent in people living in rural compared to urban geographical areas [15, 16]. As such, there is a need for contemporary data in Australia to determine the extent and sources of disparities in age-specific risk of visual and hearing impairment between these populations.

This study will be the second nationwide population-based survey on eye health and first nationwide population-based survey on ear health in Australia. The results of the AEEHS will provide valuable information regarding Australia's current eye and ear health status, outlining gaps in treatment, diagnosis and prevention, and assessing potential links between eye disease, hearing loss and critical health and/or social outcomes. This will assist in developing public health policy initiatives, guiding future resource allocation and delivery of vision and hearing services to better prevent and manage vision and hearing loss. Furthermore, it will be possible to track changes in eye health since the first National Eye Health Survey (NEHS1) in 2016 to assess whether progress has been made to lessen the burden of vision loss.

The key objectives of this study are:

- To establish the current prevalence, causes, risk factors and impacts of visual and hearing impairment in Australian adults.

- To identify the modifiable and non-modifiable risk factors for vision and hearing loss in adults.

- To analyse trends in eye health since the 2016 NEHS1 and report key metrics for universal eye health coverage e.g. effective cataract surgery coverage, effective refractive error coverage.

- Compare the prevalence of visual and hearing loss between Indigenous and non-Indigenous Australians, and between each urbanisation level.

## Materials and methods

The Australian Eye and Ear Health Survey (AEEHS) is a cross-sectional study which aims to recruit and examine a total of 5,000 participants across 30 randomly selected sites in Australia. Recruitment for this study began on the 1st August 2022 and is expected to end on the 31st December 2024. Sample size calculation assumed a similar prevalence of visual impairment in the NEHS1.

The protocol for this study was approved by the University of Sydney Human Research Ethics Committee (HREC-2020/818) and Australian Institute of Aboriginal and Torres Strait Islander Studies (AIATSIS) Research Ethics Committee (HREC-EO303-20211008).

### Sample size

A total of 5,000 participants will be surveyed across 30 randomly selected sites in Australia. The target population includes 1,750 Indigenous Australians (aged 40 years and older) and 3,250 non-Indigenous Australians (aged 50 years and older). Aboriginal and Torres Strait Islander participants have a younger inclusion age criteria because major eye and hearing conditions such as diabetic retinopathy (DR) and glue ear have an earlier onset and faster rate of progression in these communities [17, 18]. This study is a follow-up study of the NEHS1, thus the sample sizes were determined from the NEHS1 to ensure comparability.

The sample size calculation assumed a similar prevalence of vision impairment in non-Indigenous Australians to the pooled prevalence reported in previous studies, the Melbourne Vision Impairment Project [19] and the Blue Mountains Eye Study [20] of 5.2% and a prevalence of 17.2% in Indigenous Australians, as reported in the National Indigenous Eye Health Study [21].

Assuming a margin of error of 1.1% for non-Indigenous Australians and 2% for Indigenous Australians, a design effect of 1.5 that adjusted for interclass correlations and a 20% non-response rate, the required sample size for the non-Indigenous and Indigenous populations was approximately 3000 and 1500, respectively. Based on this sample size, and an expected cluster of 100 non-Indigenous individuals and 50 Indigenous individuals in each site, 30 recruitment sites were required [22]. In order to correct for an estimated 8% non-Indigenous, and 16% Indigenous rate of ungradable images (due to cataract, poor compliance, corneal opacity etc [23]), sample size for non-Indigenous and Indigenous populations was increased to 3250 and 1750 participants respectively.

Assuming a statistical significance level of 0.05 and sampling of 1,750 Indigenous and 3250 non-Indigenous Australians as in the NEHS1, we will have a power >99% to detect the same difference in Vision Impairment prevalence that has been previously reported (gap of 4.7%, i.e. 11.2% vs 6.5% respectively), in NEHS1 findings [13]. Further, we will have 90% power to measure a narrowing of the gap in VI prevalence between the two groups, of 2.87% or wider. There will also be over 95% power to detect a linear trend of decreasing vision impairment in the non-Indigenous cohort (using the Cochran-Armitage test for trend in proportions with continuity correction). These figures assume a statistical significance level of 0.05 and are based on the reported prevalence among Indigenous and non-Indigenous participants in the first National Eye Health Survey (NEHS1) respectively. The same sample size will provide over 90% power to accurately obtain hearing impairment prevalence.

## Site selection

Multi-stage, random-cluster sampling was used to select the 30 geographic areas shown in Table 1 to provide a representative sample. The first stage of sampling involved selecting sites at the Australian Statistical Geography Standard (ASGS) Statistical Area Level 2 (SA2) areas [14]. SA2 geographic areas refer to population clusters of approximately 10,000 residents. In turn, SA2 clusters are comprised of smaller SA1 geographic units, each of which contains approximately 200–800 persons. There are 2097 SA2s and 57,523 SA1s in Australia [14].

A second level of stratification was employed based on Indigenous status. Each state and territory had the SA2 sites allocated to a high Indigenous proportion group and a low Indigenous proportion group. This was done to ensure the target of at least 1750 Indigenous and 3250 non-Indigenous persons will be achieved, due to the lower overall prevalence of Indigenous individuals. An 80th percentile cut-off (per state) was used to determine candidacy for the low and high Indigenous status group. This resulted in 12 high Indigenous proportion SA2 sites, and 18 low Indigenous proportion SA2 sites, approximating the ratio of 1750 to 3250 participants.

A third level of stratification was based on remoteness area as defined by the ASGS, in order to achieve a similar distribution of the Australian population according to whether they lived in cities or more rural locations. This determined that the SA2 sites should be distributed as 19 Major City, 7 Inner Regional, 3 Outer Regional and 1 Remote/Very Remote sites. At this stage, the number of sites per state and territory was allocated to be approximately proportional to the Australian population living in that state/territory. This was done to ensure representativeness and improve generalisability to the overall national population. Once these strata were achieved, the required number of SA2 sites by state/territory were randomly selected from randomly ordered lists of the relevant SA2s.

Within each selected SA2, a constituent SA1 or cluster of SA1s was nominated. Choice of SA1 was constrained by several feasibility criteria, primary among them was the availability of an examination site during the study period that the team would be in the state/territory.

Table 1. Proposed examination sites for the AEEHS.

| State | Allocation of SA2 sites | SA2 sites | |
|---|---|---|---|
| | | Low Indigenous Proportion group | High Indigenous Proportion group |
| New South Wales / Australian Capital Territory | 10 | • Malabar-La Perouse-Chifley<br>• Seven Hills–Toongabbie<br>• Katoomba–Leura<br>• Wentworth Falls<br>• Padstow–Revesby<br>• Monash | • Moree<br>• Coonamble<br>• Tamworth<br>• Kempsey |
| Queensland | 7 | • Margate - Woody Point<br>• Aspley<br>• Noosa Heads<br>• Redcliffe | • Manoora<br>• Palm Island<br>• Yarrabah |
| Victoria | 5 | • Keilor<br>• East Bendigo–Kennington<br>• Mornington<br>• Clarinda - Oakleigh South | • Lakes Entrance |
| Western Australia | 3 | • Mandurah–East | • Roebuck<br>• Broome |
| South Australia | 2 | • Christies Beach | • Port Augusta |
| Northern Territory | 2 | • Katherine | • Yuendumu–Anmatjere |
| Tasmania | 1 | • Montrose - Rosetta | |
| Total sites | 30 | 30 | |

Accessibility was the other main criteria, as the study equipment had to be transported by van, and the examination site had to be accessible via sealed roads. If no suitable SA1 could be secured in that SA2, the next back up SA2 for that state/territory was considered from the randomly ordered list described above.

**Inclusion criteria.**

- Non-Indigenous Australians aged ≥50 years, or Indigenous Australians aged ≥40 years;

- Australian citizen or permanent resident

- Able to provide written informed consent

- Residing within selected SA2 boundaries

## Recruitment

At each site, recruiters will first leave an information pack containing a letter and information pamphlet outlining the study and a statement that recruiters will doorknock at their residence, in each mailbox. Recruiters will then go door-to-door at each site to recruit participants in each randomly selected SA1. Recruiters use a standardised script which ensures consistency and aims to maintain a high response rate to minimise selection bias. They will screen for eligibility based on the inclusion criteria, and eligible residents will be invited to participate and those who agree will be given an appointment card with the date, time and venue. Socio-demographic information including their age, gender, date of birth, address, contact details and Indigenous status will be recorded on the online database REDCap. Each home will be approached two times and any individuals who have declined to participate will not be recontacted. Residents who were not present following both door-knock attempts will be deemed non-contactable. Additional modes of recruitment will be implemented as appropriate through discussion with community leaders in order to adhere to cultural norms. These may include word of mouth, media announcements, and community engagement.

## Data collection and testing protocol

Participant examinations will take place onsite and consists of four stations. Consenting participants will undergo general health and eye assessment at Stations 1–3, followed by an ear health assessment at Station 4. The examination protocol is summarised in Fig 1. Verbal and written feedback in the form of a report is provided to each participant with appropriate referrals to an optometrist, audiologist and/or local GP if abnormalities are present. The testing protocol will take approximately 60–90 minutes per patient.

A voluntary take-home questionnaire will be provided to all participants to complete in their own time. This questionnaire gathers additional information about the participant's self-reported vision and hearing, health, mental wellbeing and lifestyle. It includes:

- National Eye Institute: Visual Function Questionnaire [24]

- Ocular Surface Disease Index [25]

- Health Outcomes (EuroQol Group EQ-5D-5L, EQ-5D Visual Analogue Scale) [26]

- Fatigue Severity Scale [27]

- Incidental and Planned Exercise Questionnaire [28]

- Food Frequency Questionnaire

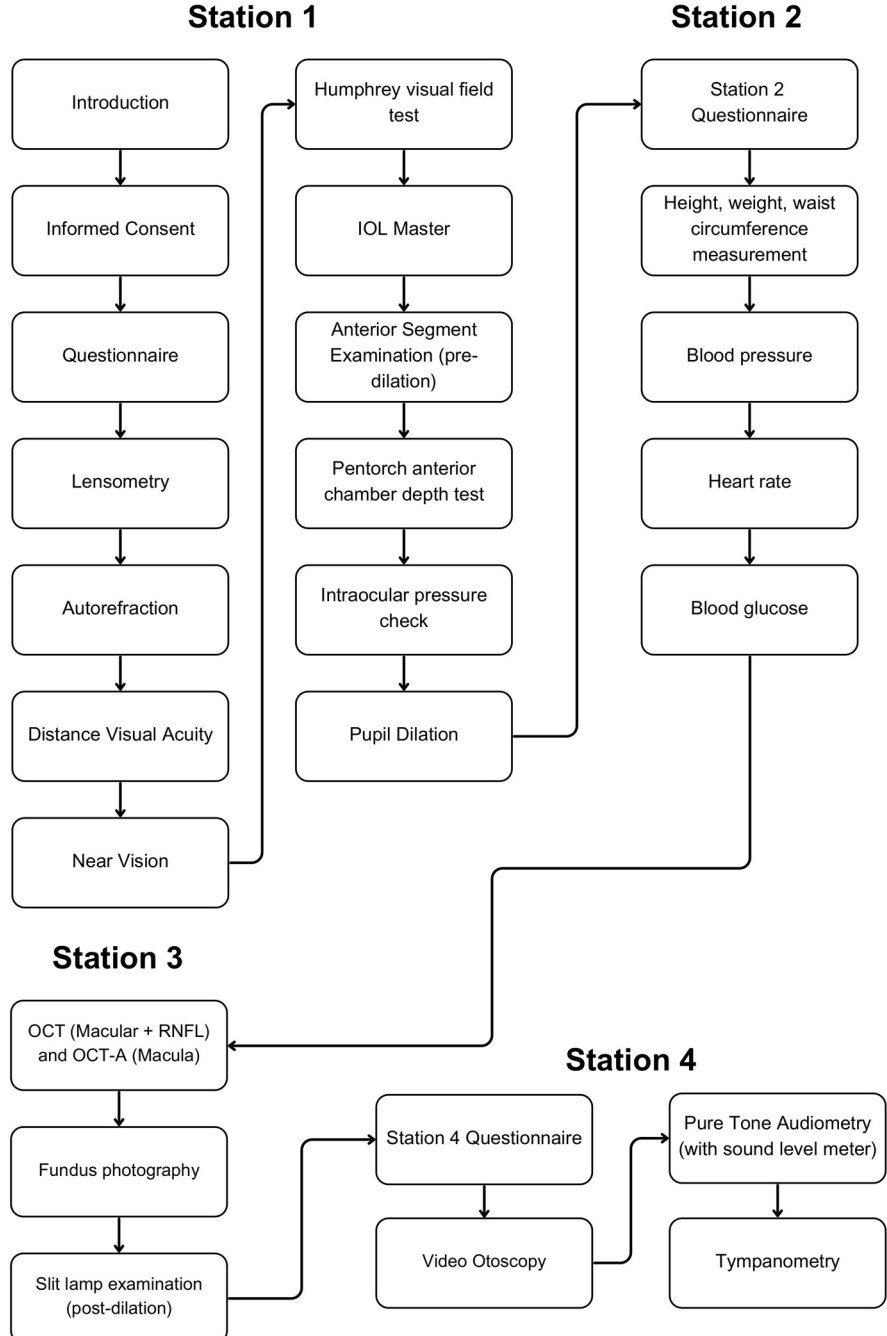

**Fig 1. Flow chart of the AEEHS examination protocol.**

- Questions about exposure to specific illnesses, noise and ototoxic chemicals, as well as history of ear infections

### Station 1

- Distance visual acuity (DVA)–Presenting DVA will be measured at 4m using calibrated, electronic ETDRS LogMAR charts in well-lit room conditions, using habitual correction with spectacles or contact lenses, if worn. If VA is less than 6/60, a Snellen chart will be used to test optotypes from 5/60 to 1/60. If no letters can be identified on the chart, VA will be assessed as counting fingers, hand movements, perception or no perception of light. Unaided DVA will also be assessed, followed by any improvement with pinhole. Vision impairment is defined as VA worse than 6/12 whereas blindness is defined VA worse than 6/60.

- Refraction–If habitual DVA ≤ 6/9.5 and improves with pinhole, DVA will be measured using trial frames with autorefraction readings as the starting point, followed by subjective refinement of sphere, cylinder and axis until best corrected visual acuity is obtained.

- Near vision acuity (NVA)–Habitual NVA is tested binocularly using a logMAR near vision chart at 40cm with spectacles, if worn. Unaided NVA will also be assessed.

Fig 2 summarises the refraction assessment performed in Station 1.

- Lensometry–the power of participants' spectacles will be measured using the Zeiss VISU-LENS 550. This provides information on refractive error coverage.

- Ocular biometry–non-contact partial coherence laser interferometry (Zeiss IOLMaster) will be used to measure axial length, anterior chamber depth, corneal curvature, lens and central corneal thickness.

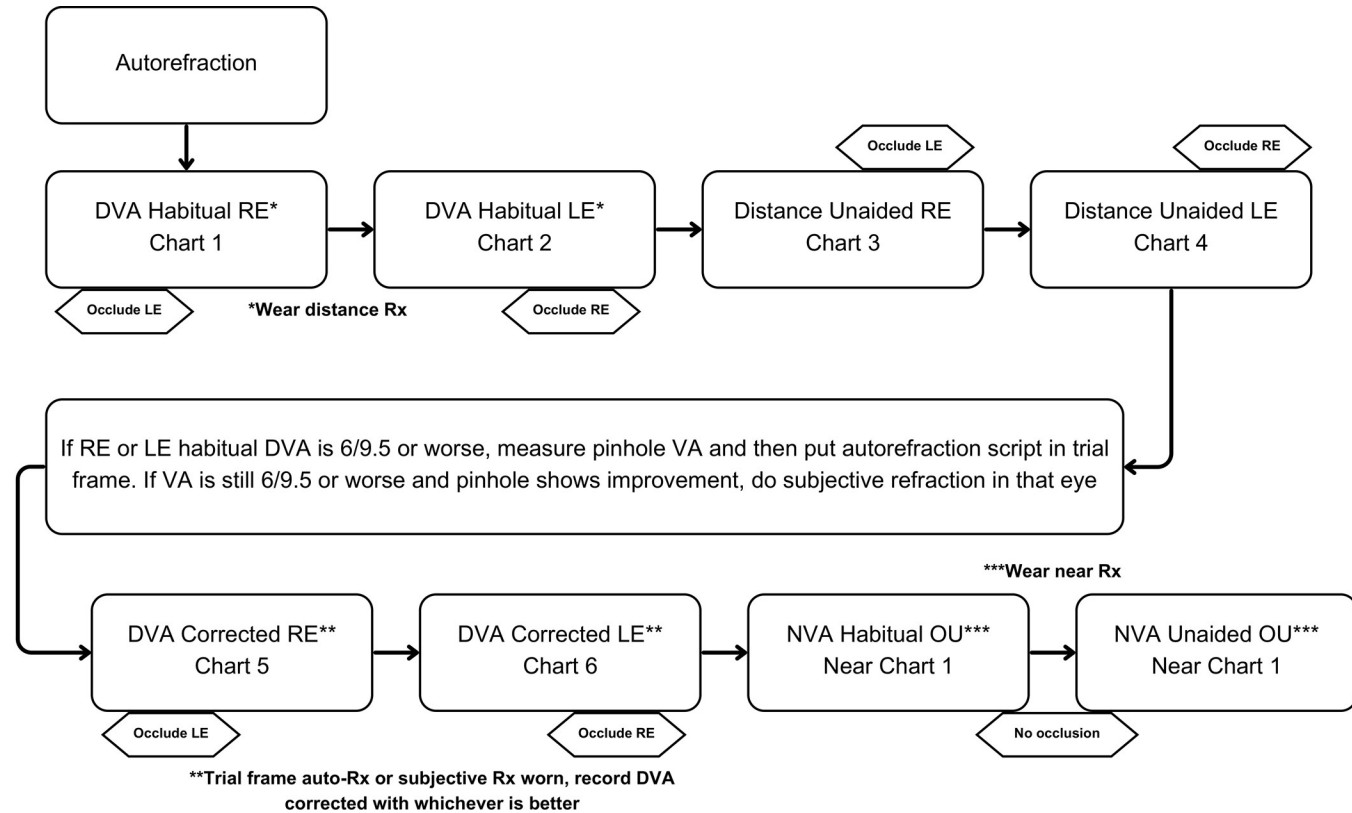

**Fig 2. Flow chart for assessment of vision and refraction.**

- Autorefraction–Zeiss VisuREF 150 autorefractor will be used to objectively measure spherical and cylindrical refractive error.

- Pre-dilation anterior segment examination–gross assessment of iris colour, pterygium, pupil, eyelid abnormalities and anterior chamber depth will be performed using a bright LED pentorch. If narrow angles are identified, then anterior segment OCT and gonioscopy will be performed on the participant.

- Tonometry–intraocular pressure (IOP) will be measured using the iCare tonometer. If IOP>25mmHg or difference between eyes ≥5mmHg, a repeat reading will be taken using Goldmann applanation tonometry (Haag-Streit).

- Visual fields–a 24–2 SITA Faster visual field examination of each eye will be performed using the Zeiss Humphrey Field Analyser III. This is used to determine visual field abnormalities which can reflect glaucoma, neurological disease and other retinal conditions.

- Pupil dilation–tropicamide 1.0% will be instilled in each eye at the end of Station 1 to dilate pupils of consenting patients to optimise eye health assessment and imaging.

## Station 2

- Questionnaire–includes questions on demographics including educational/occupational status, income, country of birth, smoking status, use of medications, and medical, surgical, ocular history (e.g. history of cataract, AMD, glaucoma, DR and previous ocular surgeries). These questions are important for linking risk factors with eye disease.

- Anthropometry–Height, weight and waist circumference will be measured using a height stand, Wedderburn electronic scale and measuring tape, respectively.

- Blood pressure and heart rate will be measured using the Omron HEM 700.

- Blood glucose will be measured using a finger prick test using the Accu-Chek glucose meter.

- Cognitive function will be measured using the Montreal cognitive assessment (MoCA) for subset of participants over 65 years.

## Station 3

- Optical coherence tomography (OCT)–This will be the first nationwide survey to use OCT imaging technology. The Zeiss Cirrus 6000 will be used to examine the optic nerve head, macula and retinal nerve fibre layer in both eyes. OCT is widely used for the diagnosis of retinal conditions including AMD, DR, macular holes and optic nerve disorders including glaucoma. Wide-field OCT-angiography scans will also be taken to non-invasively detect principal biomarkers that underlie neovascular AMD and DR. Anterior segment imaging will be used to assess anterior chamber angle on participants with Pentorch Anterior Chamber Depth Grade 1.

- Mydriatic fundus photography–Ultra-widefield colour (200˚ field of view) and fundus autofluorescence retinal images of each eye will be captured using the Zeiss Clarus 700 retinal camera. An external image of each eye will also be taken using the camera to observe the lens through the dilated pupil. Photographs will then be graded using BMES and Wisconsin grading protocols by masked, trained ophthalmologists.

- Post-dilation slit lamp examination–the slit lamp (Haag-Streit BM900) will be used to examine the anterior segment for signs of trachoma trichiasis, pseudoexfoliation, pigment dispersion, corneal opacities and cataract. The Lens Opacities Classification System III (LOCS III) [29] will be used to grade the type and severity of cataract. It consists of a series of slit-lamp images for grading nuclear, cortical and posterior subcapsular cataract. Fluorescein dye is instilled last to assess tear break-up time and corneal staining using the Oxford staining score [30] in order to detect dry eye disease.

## Station 4

- Questionnaire–includes an interviewer-administered assessment on ontological history, access to services for their hearing loss, and use of hearing devices.

- Hearing Handicap Inventory for the Elderly–questionnaire assessing self-perceived hearing handicap in participants with pre-existing hearing loss.

- International Outcome Inventory for Hearing Aids–questionnaire evaluating outcomes of aural rehabilitation in participants who utilise amplification devices

- Video Otoscopy–visual inspection of the ear canal and tympanic membrane will be performed using the MedRx video otoscope. Observations are categorised as normal or abnormal tympanic membrane and ear canal presentations. Digital photographs will be taken for each ear and graded by masked hearing professionals.

- Pure-tone audiometry–will be conducted using the Advant A2D Audiometer with passive noise reducing headphones (DD65v2) using a Hughson-Westlake staircase procedure. Noise levels will be monitored using a Bruel and Kjaer type 2250 sound level meter. Hearing loss will be determined as the pure-tone average of audiometric hearing thresholds at 500, 1000, 2000 and 4000Hz, and hearing loss will be defined as hearing thresholds >20dB hearing level.

- Tympanometry–will be undertaken using the Amplivox Otowave 102–1. Results will record numerical data including static compliance, gradient, tympanometric peak pressure and ear canal volume (ECV). Responses will be typified using the Jerger classification system for tympanograms [31].

The study team comprises a Project Manager, with overall supervision and responsibility for study conduct, execution, hiring, training of staff and collection of high-quality data. The survey team also includes door knockers with responsibility for door knocking. This usually numbers between 2–6, depending on availability, and are hired locally and provided with 1 day training by the Project Manager or delegate. Three full time research officers with prior ophthalmic or hearing experience, supplemented by PhD students, conduct most of the examinations, under supervision of the Project Manager who also participates to ensure high quality data collection. The current research officers are optometry, orthoptic and hearing science graduates. All examination staff undergo 2 weeks of training in good research conduct, familiarity with the Study Protocols, training in visual acuity and subjective refraction measurement, use of the imaging equipment, storage and backup of data, and emphasis on high quality data collection and maintaining participant confidentiality. Data collection is reviewed monthly by the study Biostatistician to identify missing data or inaccuracies, and remedial training and feedback is provided to the relevant staff member if this is identified. This regular oversight ensures consistency among data collection and examination skills of research

officers. In cases of disagreement, arbitration is provided by the Study Principal Investigators with clinical experience (PM, GL).

## Data storage

All data will be de-identified after data collection. The de-identified data collected for analysis will be stored via University of Sydney Research Electronic Data Capture (REDCap), a secure, web-based application for managing online databases. Imaging data will be stored on a secure online server at Westmead Institute of Medical Research.

Data entry, checking and cleaning are conducted in 2 stages. Initially at the end of each examination day, the Project Manager reviews data collected to ensure no missing data or anomalous values are recorded. If any issues are identified, these can be rectified on the same day. Every month, or more frequently if necessary, the Biostatistician reviews the data collected to check for consistency and any issues detected are immediately raised and corrected.

Data collection for most stations is done via entry into an electronic tablet or computer, directly into the database. Real time oversight can thus be provided by the Project Manager and Biostatistician.

## Data analysis

95% CIs for age-standardised prevalence of diseases will be calculated using the normal approximation with Australian population using 2021 Census data. ANOVA will be used to compare the mean among groups of normally distributed parameters. Chi-square tests will be used to compare proportions. Multivariate logistic regression models will be used to examine the association of potential risk factors with disease. A two-tailed p-value $<0.05$ will be considered statistically significant.

## Dissemination of findings

A one-page summary of the study findings will be shared with the participants via their choice of post or email. Study completion is expected by the end of 2024 with data analysis and reports following in the first half of 2025. Findings will be disseminated through publications in peer-reviewed journals and presentations at conferences. Findings will also be widely disseminated by project partners with the aim of improving public health policy directives and equitable service delivery to prevent avoidable vision and hearing loss in Australia.

## Supporting information

**S1 Questionnaire. Inclusivity in global research.**
(DOCX)

## Author Contributions

**Conceptualization:** Richard Kha, Paul Mitchell, Gerald Liew, Lisa Keay, Colina Waddell, Bamini Gopinath.

**Data curation:** Paul Mitchell, Vu Do, Bamini Gopinath.

**Formal analysis:** Oonagh Macken, Paul Mitchell, Vu Do.

**Funding acquisition:** Paul Mitchell, Gerald Liew, Lisa Keay, Colina Waddell, Bamini Gopinath.

**Investigation:** Richard Kha, Oonagh Macken, Paul Mitchell, Eleanor Yang.

**Methodology:** Richard Kha, Paul Mitchell, Gerald Liew, Lisa Keay, Colina Waddell, Eleanor Yang, Vu Do, Tim Fricke, John Newall, Bamini Gopinath.

**Project administration:** Richard Kha, Paul Mitchell, Gerald Liew, Lisa Keay, Colina Waddell, Vu Do, Bamini Gopinath.

**Resources:** Richard Kha, Paul Mitchell, Gerald Liew, Lisa Keay, Bamini Gopinath.

**Supervision:** Richard Kha, Paul Mitchell, Gerald Liew, Lisa Keay, Colina Waddell, Tim Fricke, John Newall, Bamini Gopinath.

**Validation:** Paul Mitchell.

**Visualization:** Richard Kha, Oonagh Macken.

**Writing – original draft:** Richard Kha, Oonagh Macken.

**Writing – review & editing:** Richard Kha, Oonagh Macken, Paul Mitchell, Gerald Liew, Lisa Keay, Colina Waddell, Eleanor Yang, Vu Do, Tim Fricke, John Newall, Bamini Gopinath.

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
