## [Decision Letter · Decision Letter 0]

8 Dec 2023

PONE-D-23-19954The Australian Eye and Ear Health Survey (AEEHS): study protocol for a population-based cross-sectional studyPLOS ONE

Dear Dr. Kha,

Thank you for submitting your manuscript to PLOS ONE. After careful consideration, we feel that it has merit but does not fully meet PLOS ONE’s publication criteria as it currently stands. Therefore, we invite you to submit a revised version of the manuscript that addresses the points raised during the review process.

The manuscript has been evaluated by two reviewers, and their comments are available below.

The reviewers have raised a number of specific concerns. They feel the manuscript should add additional details to the methods section, in particular the manuscript would benefit from a clearer explanation of sample size, inclusion and exclusion criteria and if possible, an inclusion of additional measures.

Could you please carefully revise the manuscript to address all comments raised?

We look forward to receiving your revised manuscript.

Kind regards,

Johanna Pruller, Ph.D.

Associate Editor

PLOS ONE

Journal Requirements:

Reviewers' comments:

Reviewer's Responses to Questions

**Comments to the Author**

1. Does the manuscript provide a valid rationale for the proposed study, with clearly identified and justified research questions?

Reviewer #1: Yes

Reviewer #2: Yes

2. Is the protocol technically sound and planned in a manner that will lead to a meaningful outcome and allow testing the stated hypotheses?

Reviewer #1: Partly

Reviewer #2: Yes

3. Is the methodology feasible and described in sufficient detail to allow the work to be replicable?

Reviewer #1: No

Reviewer #2: Yes

4. Have the authors described where all data underlying the findings will be made available when the study is complete?

Reviewer #1: Yes

Reviewer #2: Yes

5. Is the manuscript presented in an intelligible fashion and written in standard English?

Reviewer #1: Yes

Reviewer #2: Yes

6. Review Comments to the Author

You may also provide optional suggestions and comments to authors that they might find helpful in planning their study.

Reviewer #1: The Australia Eye Survey

Comments

General

The survey protocol is interesting to read. One important argument is, that the findings from the survey of 5,000 individuals will represent the whole of Australia, which has a population of nearly 26 million. Although the estimation of sample size was done scientifically along the scientific way of sampling, the readers will be skeptical about the representativeness and generalizability of the findings. Maybe the authors can share their justification and thoughts on this. This will help many researchers.

Specific

There is a lack of clarity in the sample size estimation.

The author can share the parameters used in the estimation of sample size from the previous survey. This will make clarity to the readers.

Again the author mentioned that….. “to detect a 2.87% difference in visual impairment prevalence….” Was the sample size estimation done based on the difference in the prevalence of visual impairment between the previous and current surveys? The author made an assumption that keeping 2.87% of the difference in the prevalence, what should be a sample size with 99% power? Needs clarification.

The author can give reasons why indigenous were being oversampled (3.2%). When one sub-population of the whole being sampled, the sample may not be representative and findings may not be generalizable. How did the author decide 3250 non-indigenous and 1750 individuals from Indigenous?

Regarding site selection, the author can explain more about SA 1 and SA 2. How many SA1 and SA2 are in the whole of Australia? Maybe the author can incorporate a flow chart of Multi-stage cluster random sampling. The author can mention clearly what cluster size and numbers to be selected for each non-indigenous and indigenous Australian.

The cluster size was 100 for non-indigenous and 50 for indigenous, to be selected from each of 30, then the numbers did not reach the estimated sample number.

First stratification is done based on state and territory, how many states and territories are in Australia? How did the number of sites being decided? (E.g., New South Wales, 6 from low and 4 from high in Table 1). This can be shared.

The author can share details of the survey team and their responsibilities. It is worth highlighting the training undergone by the team. Whether the authors plan for any estimation of agreement among teams in terms of examination skills.

Data management and storage

Maybe the author can highlight how data entry, checking, and cleaning will be done. I think data collection is planned in physical paper format, not e-format. This part is worth mentioning.

The author can also share the expected key outcomes of the survey which you think are the most important.

Reviewer #2: The paper is well written. I only have few minor comments:

1. In the introduction, 2nd para, pls ensure that the economic cost for VI and HI are correct as HI is >3 times that of VI and also hearing aids are way more expensive than VI treatment esp. cataract sx and UCRE.

2. The study will track changes in eye health with NEHS1. Would be good to clarify if the protocols are aligned between 2 studies for comparison.

3. For SS calculation, was prevalence of HI also taken into consideration? If not, why? Pls clarify.

4. Under site selection (page 6), need reference for eye diseases in Aboriginal and Torres.

5. Were people with cognitive impairment/dementia excluded as they may be unable to provide consent?

6. How about blind/deaf? Were they excluded too?

7. Recruitment: Each home will be viisted twice. Any particular reason for visiting 2 times only? Usual practice is to approach at least 4 times before they are deemed uncontactable to ensure maximum participation and good response rate.

8. A voluntary take home interview will be provided....why voluntary? Wouldn't it be better to administer on-site or make it mandate to get critical/rich data? Any previous experience of voluntary take-home uptake?

9. Why was DVA checked at 2 and not 4 m?

10. Gonioscopy is important for glaucoma diagnosis, but not included, why?

11. I don't think this study will be the first study using OCTA on nationally representative sample. Pls cross check

12. Fundus photo done undilated or dilated, pls clarify.

13. Pls be consistent with the use of acronyms e.g. AMD/ DR etc. in the manuscript.

14. What cut-offs will be used to define HI?

15. Suggest including anthropometric measures, BP/HR before ocular tests to know any contraindication before dilation.

7. PLOS authors have the option to publish the peer review history of their article (what does this mean?). If published, this will include your full peer review and any attached files.

Reviewer #1: **Yes: **Suraj Singh Senjam

Reviewer #2: No

---

## [Author Response · Author response to Decision Letter 0]

22 Feb 2024

Reviewer 1 Comments

Comment 1. The survey protocol is interesting to read. One important argument is that the findings from the survey of 5,000 individuals will represent the whole of Australia, which has a population of nearly 26 million. Although the estimation of sample size was done scientifically along the scientific way of sampling, the readers will be skeptical about the representativeness and generalizability of the findings. Maybe the authors can share their justification and thoughts on this. This will help many researchers. 

Response 1. We acknowledge the reviewers’ comment but would like to respectfully disagree. Approximately 5000 individuals, carefully selected to be representative, is the usual sample size for national population studies with similar aims. For example, the highly regarded National Health and Nutrition Examination Survey (NHANES), run by the Centers for Disease Control, USA, has been conducting regular surveys of the US population since 1960s. It recruits 5000 participants from 15 counties, and the data is accepted to be representative of the entire US population of 330 million. (https://www.cdc.gov/nchs/nhanes/about_nhanes.htm) 

The 1st National Eye Health Survey (NEHS1), which is the predecessor to our current survey, also surveyed almost 5000 participants from 30 sites, using a similar methodology, and the results are accepted to be generalizable to the wider Australian population. We are confident the results will be representative and generalizable to the wider Australian population.1,2 

1 Foreman J, Xie J, Keel S, van Wijngaarden P, Sandhu SS, Ang GS, Fan Gaskin J, Crowston J, Bourne R, Taylor HR, Dirani M. The Prevalence and Causes of Vision Loss in Indigenous and Non-Indigenous Australians: The National Eye Health Survey. Ophthalmology. 2017 Dec;124(12):1743-1752. doi: 10.1016/j.ophtha.2017.06.001. Epub 2017 Jul 6. PMID: 28689897. 

2 Keel S, Xie J, Foreman J, van Wijngaarden P, Taylor HR, Dirani M. Prevalence of Age-Related Macular Degeneration in Australia: The Australian National Eye Health Survey. JAMA Ophthalmol. 2017 Nov 1;135(11):1242-1249. doi: 10.1001/jamaophthalmol.2017.4182. PMID: 29049463; PMCID: PMC5710385.

Comment 2. There is a lack of clarity in the sample size estimation. The author can share the parameters used in the estimation of sample size from the previous survey. This will make clarity to the readers. Again the author mentioned that….. “to detect a 2.87% difference in visual impairment prevalence….” Was the sample size estimation done based on the difference in the prevalence of visual impairment between the previous and current surveys? The author made an assumption that keeping 2.87% of the difference in the prevalence, what should be a sample size with 99% power? Needs clarification.

Response 2. We apologise for the lack of detail and unclear phrasing, we now provide more detail which will hopefully address the reviewer’s concerns. 

Page 6, Lines 11-16. “Assuming a statistical significance level of 0.05 and sampling of 1,750 Indigenous and 3250 non-Indigenous Australians as in the NEHS1, we will have a power >99% to detect the same difference in Vision Impairment prevalence that has been previously reported (gap of 4.7%, i.e. 11.2% vs 6.5% respectively), in NEHS1 findings [13]. Further, we will have 90% power to measure a narrowing of the gap in Vision Impairment prevalence between the two groups of 2.87% or wider.”

Comment 3. The author can give reasons why indigenous were being oversampled (3.2%). When one sub-population of the whole being sampled, the sample may not be representative and findings may not be generalizable. How did the author decide 3250 non-indigenous and 1750 individuals from Indigenous? 

Response 3. We thank the reviewer for this important comment and the opportunity to provide further details. We now describe this in further detail. Our power calculations below indicated a minimum number of 1500 Indigenous participants needed to be sampled, in order to detect a 2% margin of error in the estimated vision impairment prevalence of 17.2%. This was increased to 1750 to account for ungradable images. Although technically an oversampling, this was driven by the power calculations, and needed to obtain reliable estimates with narrow confidence intervals. We now realise that describing this as ‘oversampling’ confuses the issue and have now deleted this sentence. 

Page 5, Line 19 to Page 6, Line 9. “This study is a follow-up study of the NEHS1, thus the sample sizes were determined from the NEHS1 to ensure comparability. 

The sample size calculation assumed a similar prevalence of vision impairment in non-Indigenous Australians to the pooled prevalence reported in previous studies, the Melbourne Vision Impairment Project [19] and the Blue Mountains Eye Study [20] of 5.2% and a prevalence of 17.2% in Indigenous Australians, as reported in the National Indigenous Eye Health Study [21].

Assuming a margin of error of 1.1% for non-Indigenous Australians and 2% for Indigenous Australians, a design effect of 1.5 that adjusted for interclass correlations and a 20% non-response rate, the required sample size for the non-Indigenous and Indigenous populations was approximately 3000 and 1500, respectively. Based on this sample size, and an expected cluster of 100 non-Indigenous individuals and 50 Indigenous individuals in each site, 30 recruitment sites were required [22]. In order to correct for an estimated 8% non-Indigenous, and 16% Indigenous rate of ungradable images (due to cataract, poor compliance, corneal opacity etc [23]), sample size for non-Indigenous and Indigenous populations was increased to 3250 and 1750 participants respectively.” 

Comment 4. Regarding site selection, the author can explain more about SA 1 and SA 2. How many SA1 and SA2 are in the whole of Australia? Maybe the author can incorporate a flow chart of multi-stage cluster random sampling. The author can mention clearly what cluster size and numbers to be selected for each non-indigenous and indigenous Australian. The cluster size was 100 for non-indigenous and 50 for indigenous, to be selected from each of 30, then the numbers did not reach the estimated sample number. First stratification is done based on state and territory, how many states and territories are in Australia? How did the number of sites being decided? (E.g., New South Wales, 6 from low and 4 from high in Table 1). This can be shared. 

Response 4. Thank you. Please see response above for explanation of how cluster sizes of 100 and 50 were obtained, and how this was increased to reach the total sample of 5000. 

The other comments are addressed below, in the revised manuscript. 

Page 7, Line 1 to Page 8, Line 7. “Multi-stage, random-cluster sampling was used to select 30 geographic areas to provide a representative sample. The first stage of sampling involved selecting sites at the Australian Statistical Geography Standard (ASGS) Statistical Area Level 2 (SA2) areas [14]. SA2 geographic areas refer to population clusters of approximately 10,000 residents. In turn, SA2 clusters are comprised of smaller SA1 geographic units, each of which contains approximately 200–800 persons. There are 2097 SA2s and 57,523 SA1s in Australia [14]. 

A second level of stratification was employed based on Indigenous status. Each state and territory had the SA2 sites allocated to a high Indigenous proportion group and a low Indigenous proportion group. This was done to ensure the target of at least 1750 Indigenous and 3250 non-Indigenous persons will be achieved, due to the lower overall prevalance of Indigenous individuals. An 80th percentile cut-off (per state) was used to determine candidacy for the low and high Indigenous status group. This resulted in 12 high Indigenous proportion SA2 sites, and 18 low Indigenous proportion SA2 sites, approximating the ratio of 1750 to 3250 participants. 

A third level of stratification was based on remoteness area as defined by the ASGS, in order to achieve a similar distribution of the Australian population according to whether they lived in cities or more rural locations. This determined that the SA2 sites should be distributed as 19 Major City, 7 Inner Regional, 3 Outer Regional and 1 Remote/Very Remote sites. At this stage, the number of sites per state and territory was allocated to be approximately proportional to the Australian population living in that state/territory. This was done to ensure representativeness and improve generalisability to the overall national population. Once these strata were achieved, the required number of SA2 sites by state/territory were randomly selected from randomly ordered lists of the relevant SA2s. 

Within each selected SA2, a constituent SA1 or cluster of SA1s was nominated. Choice of SA1 was constrained by several feasibility criteria, primary among them was the availability of an examination site during the study period that the team would be in the state/territory. Accessibility was the other main criteria, as the study equipment had to be transported by van, and the examination site had to be accessible via sealed roads. If no suitable SA1 could be secured in that SA2, the next back up SA2 for that state/territory was considered from the randomly ordered list described above.” 

Comment 5. The author can share details of the survey team and their responsibilities. It is worth highlighting the training undergone by the team. Whether the authors plan for any estimation of agreement among teams in terms of examination skills. 

Response 5. Thank you for the comment. We have added a section in the methods to detail the survey team and their responsibilities.

Page 15, Lines 1-17. “The study team comprises a Project Manager, with overall supervision and responsibility for study conduct, execution, hiring, training of staff and collection of high-quality data. The survey team comprises door knockers with responsibility for door knocking. This usually numbers between 2-6, depending on availability, and are hired locally and provided with 1 day training by the Project Manager or delegate. Three full time research officers with prior ophthalmic or hearing experience, supplemented by PhD students, conduct most of the examinations, under supervision of the Project Manager who also participates to ensure high quality data collection. The current research officers are optometry, orthoptic (ophthalmic nursing) and hearing science graduates. All examination staff undergo 2 weeks of training in good research conduct, familiarity with the Study Protocols, training in visual acuity and subjective refraction measurement, use of the imaging equipment, storage and backup of data, and emphasis on high quality data collection and maintaining participant confidentiality. Data collection is reviewed monthly by the study Biostatistician to identify missing data or inaccuracies, and remedial training and feedback is provided to the relevant staff member if this is identified. This regular oversight ensures consistency among data collection and examination skills of research officers. In cases of disagreement, arbitration is provided by the Study Principal Investigators with clinical experience (PM, GL).” 

Comment 6. Maybe the author can highlight how data entry, checking, and cleaning will be done. I think data collection is planned in physical paper format, not e-format. This part is worth mentioning.

Response 6. Thank you for the comment. We have expanded the data storage section in the manuscript.

Page 15, Line 24 to Page 16, Line 7. “Data entry, checking and cleaning are conducted in 2 stages. Initially at the end of each examination day, the Project Manager reviews data collected to ensure no missing data or anomalous values are recorded. If any issues are identified, these can be rectified on the same day. Every month, or more frequently if necessary, the Biostatistician reviews the data collected to check for consistency and any issues detected are immediately raised and corrected. 

Data collection for most stations is done via entry into an electronic tablet or computer, directly into the database. Real time oversight can thus be provided by the Project Manager and Biostatistician.” 

Comment 7. The author can also share the expected key outcomes of the survey which you think are the most important. 

Response 7. Thank you for the comment. We have now added the expected key outcomes of the survey in the Introduction section.

Page 4, Lines 13-22. “The key objectives of this study are:

- To establish the current prevalence, causes, risk factors and impacts of visual and hearing impairment in Australian adults.

- To identify the modifiable and non-modifiable risk factors for vision and hearing loss in adults.

- To analyse trends in eye health since the 2016 NEHS1 and report key metrics for universal eye health coverage e.g. effective cataract surgery coverage, effective refractive error coverage.

- Compare the prevalence of visual and hearing loss between Indigenous and non-Indigenous Australians, and between each urbanisation level.”

Reviewer 2 Comments

Comment 1. In the introduction, 2nd para, pls ensure that the economic cost for VI and HI are correct as HI is >3 times that of VI and also hearing aids are way more expensive than VI treatment esp. cataract sx and UCRE.

Response 1. Thank you for the comment. This has been changed to a more recent reference which has an updated estimate of the cost of hearing impairment in Australia.

Page 3, Lines 12-13. “In Australia, the economic impact of vision and hearing loss is $27.6 and $33.3 billion per year, respectively [3, 5-7].”

Comment 2. The study will track changes in eye health with NEHS1. Would be good to clarify if the protocols are aligned between 2 studies for comparison.

Response 2. Thank you for the comment. The sample size of 3250 non-Indigenous and 1750 Indigenous Australians remains the same as NEHS1. However, some of the protocols are different because this survey aims to improve upon the first survey by using new and more objective, comprehensive testing methods such as OCT, ilated fundus photography and Humphrey visual field testing, which was not performed in the NEHS1. We will also obtain additional data by using the autorefractor, IOLMaster and more comprehensive questionnaires to capture critical data. Protocols for NEHS1 for similar outcomes such as visual acuity were followed and expanded upon e.g. subjective refraction was also performed in this survey, to ensure comparability while also obtaining extra data. 

Comment 3. For SS calculation, was prevalence of HI also taken into consideration? If not, why? Pls clarify.

Response 3. Thank you for the comment. The prevalence of hearing impairment was not taken into account in the original sample size calculation as the hearing component of the study was added after the original eye health survey was funded. This occurred because the investigators were able to secure additional funding through other sources to conduct the hearing components of the study. The investigators have previously conducted combined vision and hearing studies, and this was a natural extension to the current eye survey.3,4

3 Gopinath B, McMahon CM, Burlutsky G, Mitchell P. Hearing and vision impairment and the 5-year incidence of falls in older adults. Age Ageing. 2016 May;45(3):409-14. doi: 10.1093/ageing/afw022. Epub 2016 Mar 5. PMID: 26946051.

4 Schneider J, Gopinath B, McMahon C, Teber E, Leeder SR, Wang JJ, Mitchell P. Prevalence and 5-year incidence of dual sensory impairment in an older Australian population. Ann Epidemiol. 2012 Apr;22(4):295-301. doi: 10.1016/j.annepidem.2012.02.004. Epub 2012 Mar 3. PMID: 22382082.. 

Comment 4. Under site selection (page 6), need reference for eye diseases in Aboriginal and Torres.

Response 4. Thank you for the comment. References have been added for the statement regarding Aboriginal and Torres Strait Islander people in the aforementioned section.

Page 6, Lines 10-13. Aboriginal and Torres Strait Islander participants have a younger inclusion age criteria because major eye and hearing conditions such as diabetic retinopathy (DR) and glue ear have an earlier onset and faster rat

---

## [Editor Report · Decision Letter 1]

24 Mar 2024

The Australian Eye and Ear Health Survey (AEEHS): study protocol for a population-based cross-sectional study

PONE-D-23-19954R1

Dear Dr. Kha,

We’re pleased to inform you that your manuscript has been judged scientifically suitable for publication and will be formally accepted for publication once it meets all outstanding technical requirements.

Kind regards,

Suraj Singh Senjam, MD, MSc PH for Eye Care

Guest Editor

PLOS ONE

Additional Editor Comments (optional):

Comments

Thank you to all the authors for the revision that was done nicely, especially the sample size estimation in each survey site.

I have a few minor comments now.

On page 15, lines 1 and 3, “the study team comprises” was used two times. The author can paraphrase it.

NOTE:

(The following is the argument among us, within scientific communities, not to be included in the manuscript)

Regarding the issue of sample size whether it is representative of a country or not, my suggestion or personal experience is even if we used all robust scientific methods for the sample size estimation, for advocacy purposes, the decision or policymakers usually considered a study with a bigger sample size. For example, hypothetically a separate team conducts a similar or the same study with the same objectives with a bigger sample size (N=15000) using the same sampling technique, the policymaker prefers the study with the bigger sample size to be used for the planning and policy purposes. Additionally, our goal of population-based study is for advocacy purposes.

Even “face validity on the representativeness” also, the study with a bigger sample seems more valuable.

Therefore, I feel sometimes that face validity matters.
---

## [Editor Report · Acceptance letter]

8 Apr 2024

PONE-D-23-19954R1 

PLOS ONE

Dear Dr. Kha, 

I'm pleased to inform you that your manuscript has been deemed suitable for publication in PLOS ONE. Congratulations! Your manuscript is now being handed over to our production team.

Kind regards, 

on behalf of

Dr. Suraj Singh Senjam 

Guest Editor

PLOS ONE